# Past, Present, and a Glance into the Future of Multiple Myeloma Treatment

**DOI:** 10.3390/ph16030415

**Published:** 2023-03-08

**Authors:** Weam Othman Elbezanti, Kishore B. Challagundla, Subash C. Jonnalagadda, Tulin Budak-Alpdogan, Manoj K. Pandey

**Affiliations:** 1Department of Biomedical Sciences, Cooper Medical School of Rowan University, 401S Broadway, Camden, NJ 08103, USA; 2Department of Hematology, MD Anderson Cancer Center at Cooper, Cooper Health System, Camden, NJ 08103, USA; 3Department of Biochemistry and Molecular Biology, The Fred and Pamela Buffett Cancer Center, The Child Health Research Institute, University of Nebraska Medical Center, Omaha, NE 68198, USA; 4Department of Chemistry and Biochemistry, Rowan University, Glassboro, NJ 08028, USA

**Keywords:** multiple myeloma, B-cell malignancy, bone marrow microenvironment, drug resistance, proteasome inhibitors, immunotherapies

## Abstract

Multiple myeloma (MM) is a challenging hematological cancer which typically grows in bone marrow. MM accounts for 10% of hematological malignancies and 1.8% of cancers. The recent treatment strategies have significantly improved progression-free survival for MM patients in the last decade; however, a relapse for most MM patients is inevitable. In this review we discuss current treatment, important pathways for proliferation, survival, immune suppression, and resistance that could be targeted for future treatments.

## 1. The Pathogenesis of Multiple Myeloma

Multiple myeloma (MM) is a mature B-cell neoplasm that is characterized by uncontrolled growth of plasma cells (PCs) in bone marrow (BM) which leads to excessive secretion of antibodies. The progression of MM is a multistep process that starts with an asymptomatic premalignant condition known as monoclonal gammopathy of undetermined significance (MGUS), in which BM produces abnormal PCs and secretes M protein instead of normal antibodies [1]. With the increase in oncogenic mutations, MGUS evolves into smoldering MM (SMM), which is characterized by a higher serum level of M protein and a higher percentage of clonal PCs. About 50% of patients with SMM show a constant increase of M protein and develop MM [2]. While both MGUS and SMM are asymptomatic, complications of accumulated proteins may start to affect the kidneys [3]. Almost 20–40% of MM patients have renal disease by the time of diagnosis [4]. 

The complexity of MM is attributed to the clinical and biological heterogeneity of the disease that further genetically evolves during its progression [5]. MM cells have a wide range of genetic changes including point mutations, insertions, deletions, multiploidy, and chromosomal translocations [6]. For example, trisomic MM and patients with t(11;14) are considered standard-risk patients. On the other hand, MM patients with t(4;14), t(14;16), t(14;20), p53 mutation, gain 1q, or del(17p) are considered to be high-risk [7,8]. Moreover, the bone marrow microenvironment (BMM) plays an important role in disease development, progression, and resistance [9]. All these factors enhance different signaling pathways that contribute to proliferation, survival, invasion, angiogenesis, and osteoclastogenesis [10]. There are many signaling pathways that protect against apoptosis and support MM growth which become activated through the adhesion of MM to the BMM. These activated pathways include phosphatidylinositol-3-kinase (PI3K)/protein kinase B (AKT)/mammalian target of rapamycin (mTOR), and nuclear factor kappa B (NF-κB), janus kinase 2 (JAK2)/signal transducer and activator of transcription 3 (STAT3), which support MM growth and protect against apoptosis (Figure 1). The activation of these pathways leads to upregulation and secretion of several cytokines and factors from both MM and BMM cells such as interleukin-6 (IL-6), insulin-like growth factor-1, VEGF, tumor necrosis factor alpha (TNF-α), and transforming growth factor-β [11,12]. 

The BMM contains several specialized cells that are responsible for skeletal integrity, immunity, and blood formation [13,14] (Figure 2). Its compartments are classified into niches: the immune niche, the vascular niche and the endosteal niche [15]. Each niche contains various cell types such as B-cells, T-cells, myeloid-derived suppressor cells, osteoclasts, natural killer (NK) cells, mesenchymal stem cells, BM stromal cells (BMSCs), osteoblasts, and endothelial progenitor cells [16].

BMSCs are a heterogeneous cell population that supports hematopoiesis in normal conditions. They play an important role in supporting the survival, proliferation, and drug resistance of MM [17]. They communicate with MM in several ways. The direct cell adhesion contact through adhesion molecules such as very late antigen 4 (VLA-4), vascular cell adhesion protein, lymphocyte function-associated antigen-1, and intercellular adhesion molecule 1 stimulates IL-6 secretion by BMSCs [18] and mediates drug resistance through cellular-adhesion-mediated drug resistance (CAM-DR) [19]. Stromal cell-derived factor-1α (SDF-1α), a chemokine produced by BMSCs, plays an important role in embryogenesis, angiogenesis, hematopoiesis, and inflammation [20,21]. It stimulates homing and migration of cells through G protein-coupled receptor C-X-C chemokine receptor type 4 (CXCR4) [22]. The SDF-1α/CXCR4 axis plays an important role in survival, angiogenesis, metastasis, invasion, and adhesion in MM (Figure 2) [23]. It has been shown that the SDF-1α level in MM is elevated and this elevation contributes to activating several signaling pathways and induces mitogen-activated protein kinase kinase1/2, AKT phosphorylation, mitogen-activated protein kinase (MAPK), and NF-κB in MM cell lines and patient samples [24,25]. The SDF-1α/CXCR4 axis mediates drug resistance through different pathways that are involved in CAM-DR, affecting adhesion molecules, enhancing IL-6 mediated drug resistance, and stimulating pathways including MAP/extracellular signal-regulated kinase (ERK), wingless/integrated3 (Wnt3)/Ras homolog family member A/Ras homologous -associated protein kinase, and Ras homologous/Ras homologous -kinase [26,27,28,29,30].

Other cytokines and growth factors secreted from BMSCs suppress the immune response and facilitate MM immune evasion. One of the most important cytokines in MM is IL-6, which is secreted by different BM cells including BMSCs, osteoclasts, and macrophages [31]. IL-6 secretion stimulates the JAK/STAT3 pathway, which leads to increased survival and proliferation through the upregulation of Mcl-1, Bcl-xL, Bcl-2, c-Myc, and cyclin D1 [32,33,34,35]. As MM progresses, osteoclast activity increases, which in turn causes bone lesions. During disease progression, an imbalance occurs in receptor activator of NF-κB ligand (RANKL), and osteoprotegerin [36]. 

In autologous hematopoietic progenitor cell (HPC) transplantation, plerixafor, a specific antagonist of SDF-1α binding to CXCR4, was approved in 2008 to induce hematopoietic stem cells (HSCs) and progenitor cells (HPCs) trafficking. It has been shown that it augments granulocyte colony-stimulating factor (G-CSF)-induced mobilization of HSCs and HPCs [37,38].

## 2. The Existing Therapies for MM

In the 1960s, oral melphalan, an alkylating agent, in combination with prednisone was considered the frontline treatment for MM [39,40]. Then, FDA-approved thalidomide, an immunomodulatory agent (IMiDs), was introduced in MM therapy. Thalidomide enhanced the overall survival (OS) and showed longer progression-free survival (PFS) regardless of patient age when used in combination with melphalan and prednisolone (clinical trial # NCT00232934, and ISRCTN90692740) [41,42]. Additionally in the 1980s, autologous stem cell transplantation (ASCT) followed by a high dose of therapy was introduced and became the standard of care among younger patients with normal renal function [43,44].

The discovery and the introduction of proteasome inhibitors (PIs) in 2003 has tremendously improved PFS in patients [45]. Bortezomib became the first line of treatment for MM in newly diagnosed MM (NDMM) patients. For relapsed and refractory MM (RRMM) patients, it was used in combination with melphalan and prednisone (Table 1) [46]. After the success of bortezomib, other PIs such as carfilzomib and ixazomib were approved for the MM treatment. In the TOURMALINE-MM1 trial (NCT01564537), oral ixazomib was tested in combination with lenalidomide and dexamethasone (IRd) on RRMM patients and it has significantly improved the PFS (20.6 months in the IRd group vs. 14.7 months in the Rd group (lenalidomide and dexamethasone) at a median follow-up of 14.7 months). The overall response rate (ORR) was 78% in the IRd group and 72% in the Rd group. The median OS was not reached at a median follow-up of approximately 23 months. The additional adverse effects between the two groups were limited and there was a similar quality of life between the IRd and the Rd groups [47].

The introduction of a new generation of IMiDs such as lenalidomide in 2005, in combination with PIs, increased survival from 14.8 to 30.9 months [56]. Currently, a triple therapy (PI, IMiD, and corticosteroids) is the first line of treatment for MM followed by autologous stem cell transplant (ASCT). Lenalidomide is usually recommended as a maintenance therapy for MM patients [57]. A combination of bortezomib, lenalidomide, and dexamethasone (VRd) was tested on NDMM in the SWOG-S077 phase III clinical trial (NCT00644228) versus Rd. There was a significant improvement in median PFS (43 months in VRd group vs. 30 months in Rd group) and in median OS (75 months in VRd vs. 64 months in the control group). The ORR was 82% in the VRd group vs. 72% in the Rd group [58]. The next generation IMiD, pomalidomide is shown to be effective and is one of the treatment options that is usually considered in combination after the first relapse for patients who are refractory to lenalidomide [59]. It was first approved in 2013 in combination with dexamethasone for RRMM patients [60]. Then, it was approved in combination with anti-CD38 monoclonal antibody and a steroid for RRMM patients who have previously received two therapies including lenalidomide and bortezomib [60,61,62,63]. 

Introducing daratumumab, an anti-CD38, in clinical trials (MAIA, ALCYONE, CASTOR, and POLLUX) with different combinations improved minimal residual disease negativity (MRD), and PFS [64] (Table 2). In 2019, daratumumab, lenalidomide, and dexamethasone (DRd) treatment was approved in NDMM patients who are ineligible for transplant after phase III MAIA trial (NCT02252172). In this study, DRd showed significant improvement in PFS (not reached) compared with lenalidomide and dexamethasone (Rd) (31.9 months). The median OS was not reached at a median follow-up of 56.2 months. The common adverse effects of this treatment are neutropenia, pneumonia, anemia, and lymphopenia. Treatment-related-death was 4% in the DRd group compared to 3% in the Rd control group [65,66]. In addition, daratumumab was tested in combination with bortezomib and melphalan-prednisone (D-VMP) in a phase III ALCYONE trial (NCT02195479) in NDMM patients. The 18-month PFS was 71.6%. At a median follow-up of 16.5 months, 22.3% of the patients were negative for MRD. The common adverse effects were neutropenia, thrombocytopenia, and anemia [67]. After the CASSIOPEIA phase III trial (NCT02541383) on ND transplant-eligible MM patients, daratumumab was approved to be used in combination with bortezomib, thalidomide, and dexamethasone (D-VTd) in 2019. At a median follow-up of 35.4 months, PFS was not reached versus 46.7 months with the control group. The most common adverse effects were lymphopenia, hypertension, and neutropenia [68].

Treatment choices differ according to age, cytogenic abnormalities, and eligibility for transplantation. Maintenance therapy for standard-risk MM patients is lenalidomide. However, bortezomib is used as a maintenance therapy for high-risk ND patients who are determined to be eligible for ASCT. ND high-risk patients who are eligible for ASCT start with three to four cycles of VRd or three to four cycles of quadruplet regimen of daratumumab, bortezomib, lenalidomide, and dexamethasone (DVRd) [59].

### 2.1. Mechanism of Action of Proteasome Inhibitors

PIs kill myeloma cells through different pathways (Figure 3). Inhibition of proteosomes leads to the accumulation of ubiquitinated proteins that would otherwise be degraded in the proteosome. This leads to the accumulation of these proteins in the endoplasmic reticulum (ER), which in turn causes ER stress, which leads to ER stress-dependent apoptosis and activation of the Jun amino-terminal kinases (JNKs) pathway, increasing the Fas ligand, caspase 8, and caspase 3 [82,83]. Furthermore, mitochondrial injury occurs due to the direct effect of ubiquitinated proteins and the indirect effect of the ER stress that releases reactive oxygen species (ROS) [84]. The direct apoptosis effect of PIs can also occur through accumulation and phosphorylation of P53, which stimulates pro-apoptotic proteins such as Bcl-2-associated X protein (Bax), NADPH oxidase activator (NOXA), cytochrome-c release, and inhibition of the antiapoptotic protein Mcl-1 [85,86].

Bortezomib is the first-generation FDA-approved PI that reversibly inhibits the chymotrypsin-like activity of the proteasomes [87]. Bortezomib has been shown to inhibit NF-κB, which in turn inhibits its downstream pathways and their products, including IL-6, vascular endothelial growth factor (VEGF), c-Myc, and cyclin D1 [88]. On the other hand, bortezomib has been shown to induce constitutive NF-κB activity which could be due to the difference in response among different cell clones [89,90] (reviewed in [86]). In addition, bortezomib ameliorates CAM-DR by inhibiting the expression of adhesion molecules such as VLA-4. Therefore, it resensitizes MM cells to treatment [91]. Even though introducing bortezomib has numerous benefits for patients, several side effects such as peripheral neuropathy may occur. The second-generation PI, carfilzomib, which does not cross the blood–brain barrier may have lower incidence of neuropathy. However, caution should be taken in using carfilzomib in elderly patients as it has shown cardiovascular side effects (reviewed at [92]). Similarly, ixazomib showed lower neurotoxicity than bortezomib as well as more efficacy in clinical trials [93]. 

### 2.2. Mechanism of Action of Immunomodulatory Drugs

IMiDs target some proteins for ubiquitination and proteasomal degradation through binding with cereblon ubiquitin ligase, forming an E3 ubiquitin ligase complex with DNA damage-binding protein 1, Cullin-4A, and RING box protein-1 (Figure 4). They target IKAROS family zinc finger 1 and 3 (IKZF1 and IKZF3), which are transcription factors that play an important role in lymphocyte biology. IKZF3 is an essential transcription factor in plasma cell development and therefore its degradation affects MM progression [94].

Another protein that has been shown to be degraded by IMiDs is casein kinase 1 alpha (CK1α), which plays an important role in the pathogenesis of MM. CK is one of the serine/threonine kinases that is important in cell survival and has been shown to be important in different types of cancer including MM [95]. Manni et al. have shown that CK1α is overexpressed in most patients’ samples and its inhibition leads to apoptosis, a decrease in β-catenin and AKT expression, and an increase in p53 and p21 expression. In addition, the same group showed that CK1α inhibition enhances the cytotoxicity of bortezomib and lenalidomide on MM [96]. Interestingly, both CK1α and CK2 have been shown to sustain activation of important signaling pathways such as JAK/STAT, NF-κB, and PI3K/AKT [95,97,98]. IMiDs also inhibit the proliferation of PCs by inhibiting the cyclin-dependent kinase pathway through inducing P21 [99]. Moreover, IMiDs induce direct apoptosis in PCs by activation of Fas-mediated cell death [100,101]. Moreover, Hideshima et al. showed that IMiDs inhibit the kinase activity of p53-related protein kinase (TP53RK), which correlate negatively with MM patients’ survival. TP53RK phosphorylate serine 15 of p53, which in turn affects MM growth. The binding of IMiDs to TP53RK triggers apoptosis by inducing pro-apoptotic protein Bim. It also affects the metastasis of MM cells by inhibiting c-Myc protein [102]. 

The treatment of IMiDs helps in restoring the immune homeostasis. Several mechanisms have been proposed for IMiDs-mediated immune restoration, for example thalidomide stimulates the proliferation of T-cells and increases the secretion of interferon-gamma (IFN-γ) and interleukin-2 (IL-2) [103,104]. Along these lines, lenalidomide has been shown to stimulate T-cell-mediated cytotoxicity, induction of T-cell proliferative responses to allogeneic dendritic cells, and suppresses expression of programmed cell death protein-1 (PD-1) [105,106]. In addition, lenalidomide and pomalidomide suppress forkhead box P3 transcription factor and T regulatory cell expansion [107]. Moreover, lenalidomide enhances the expression of Fas ligand on NK cells and increases granzyme secretion which correlates to an increase in Ab-dependent cellular cytotoxicity (ADCC) [100]. Lenalidomide and pomalidomide inhibit the expression of adhesion molecules and inhibit the RANKL/osteoprotegerin ratio, which leads to inhibition of osteoclast formation [108].

### 2.3. Mechanism of Action of Histone Deacetylase Inhibitors

Dysregulation in epigenetics including histone acetylation has been shown in different types of cancer including MM [109,110]. Mithraprabhu et al. have shown that class I histone deacetylase (HDAC) is significantly upregulated in MM patients’ samples compared with normal PCs [111]. Moreover, upregulation of HDAC1 was correlated with poor prognosis and shorter OS [111]. 

Removal of the acetyl group from the lysine residue on histone by HDACs causes transcription repression (Figure 5). HDACs affect different proteins via deacetylation either directly or indirectly by affecting the function of the chaperone protein that is needed for their stabilization. HDACs cause hyperacetylation and therefore destabilization for the chaperone protein heat shock protein 90 (HSP90), which inhibits its association with CXCR4 leading to proteasomal degradation of CXCR4 in acute myeloid leukemia (AML) cells [112,113]. HDACs also cause degradation of protein phosphatase 3 catalytic subunit alpha (PPP3CA), which is overexpressed in MM, and patients show poor prognosis when it is overexpressed. As HDAC6 plays an important role in the aggresomal protein degradation, its inhibition significantly synergizes with proteasomal inhibition in MM [114]. Hideshima et al. have shown that a selective HDAC6 inhibitor increased the cytotoxicity of bortezomib in vitro and overcame its resistance through JNK activation and ER stress [115]. In 2015, panobinostat was approved for treatment of RRMM patients in combination with dexamethasone. HDAC inhibitors (HDACis) have been shown to affect the acetylation of histone and non-histone proteins; they therefore affect different cell processes including apoptosis, survival, angiogenesis, and the cell cycle [116,117].

## 3. The Development of Immunotherapies in MM

Currently, there are rapid advances in immunotherapy to treat different types of cancer including MM. Several approaches for immunotherapies have been developed such as inhibiting immune check points, targeting antigens, development of antibody–drug conjugates, chimeric antigen receptor (CAR)-T cells, CAR-NK cell therapy, or using bispecific antibodies or bispecific T-cell engagers antibodies (BiTEs) to attach to more than one target [118]. It is important in immunotherapy to have a specific target, which ideally are surface proteins that are only expressed on target cells to minimize the side effects. MM cells express B-cell maturation antigen (BCMA), CD56, CD117, CD150 (or SLAMF1, signaling lymphocytic activation molecule1), CD48 (SLAMF2), CD229 (SLAMF3), CD352 (SLAMF6), CD319 (SLAMF7 or CS1), CD86, CD184, CD200, and CD272 [119,120,121,122]. BCMA, a tumor necrosis factor receptor superfamily 17 (TNFRSF17) member, is one of the most studied antigens for development of immunotherapies [123]. 

### 3.1. Monoclonal Antibodies

Monoclonal antibodies can be used to target surface markers on cancer cells or block immune checkpoints between immune and cancer cells. Daratumumab (anti-CD38) is the first monoclonal antibody that was approved to treat MM in 2015. CD38 is a type II transmembrane glycoprotein that is highly expressed in MM and other hematological malignancies, and it is expressed on myeloid and lymphoid cells [124]. It has been shown that daratumumab induces ADCC, complement-dependent cytotoxicity [125], and Ab-dependent cellular phagocytosis (ADCP) [126]. Moreover, immune profiling of daratumumab in patients’ samples from two clinical trials (NCT00574288 [GEN501] and NCT01985126 [SIRIUS]) showed that daratumumab exerts immunomodulatory effects via suppression of the immunosuppressive cells such as myeloid-derived suppressor, regulatory T-, and regulatory B-cells [127]. In addition, it enhances the expansion of T-helper cells and cytotoxic T-cells [127]. Another anti-CD38 antibody, isatuximab, has been approved for RRMM and NDMM patients, and shown a significant increase in ORR in different clinical trials (Table 3). It is being tested in quadruple therapy regimen with lenalidomide, bortezomib, and dexamethasone in a phase III clinical trial (NCT03617731) [128]. There was a significant improvement in minimal residual disease negativity (50% vs. 36% in the control group). Moreover, addition of isatuximab has significantly improved the very good partial response (VGPR): 77% vs. 61% in the control group [128].

Elotuzumab (anti-SLAMF7) was also approved in 2015 to be used in combination with lenalidomide and dexamethasone for MM patients who have received at least three prior therapies. SLAMF7 or CS1 is a type I transmembrane glycoprotein which belongs to the Ig superfamily, and it is highly expressed in MM cells (≥95% of cases of MM), normal PCs, and other immune cells such as dendritic, NK, and some T-cell subsets [142]. Elotuzumab induces ADCC by specific binding to SLAMF7 on MM through its Fab portion, and engages NK cells through binding of its FC portion with CD16 on NK cells. Therefore, NK cells become activated and release cytotoxic granules to kill MM cells and release IFN-γ to stimulate other immune cells [142]. Elotuzumab has been shown to induce ADCP through binding of its FC portion with Fc-gamma receptors on macrophages [143]. Moreover, elotuzumab inhibited soluble SLAMF7-induced growth of MM in vitro and in vivo [144]. Additionally, treatment using elotuzumab inhibits the adhesion of myeloma cells to BMSCs [142]. 

Immune checkpoints are major contributors of cancer evasion [145]. Targeting inhibitory immune checkpoints have revolutionized cancer therapy. There are several immune checkpoint inhibitors that have been shown to inhibit immune function such as cytotoxic T-lymphocyte antigen 4, T-cell immunoglobulin mucin-3, programmed cell death 1 (PD-1) or its ligand, programmed cell death ligand 1 (PD-L1, also referred to as B7-H1 or CD274) [146]. There are many mAbs targeting immune checkpoints approved by the FDA to treat different kinds of cancer as they prolong OS [147]. Tamura et al. have shown that PD-L1 is upregulated in MM and its upregulation is induced by IL-6. Moreover, PD-L1^+^ RPMI8226 cells have higher Bcl-2 and FasL expression compared with PD-L1^−^ RPMI8226 cells and PDL-1 upregulation is associated with drug resistance, higher MM cell percentages in BM, and higher serum lactate dehydrogenase levels [148]. Stromal-cell-induced MM growth was abrogated by blocking PD1/PD-L1 and enhanced in combination with lenalidomide ex vivo [149]. Durvalumab and nivolumab, PD-1 inhibitors, are being tested in combination with other compounds such as IMiDs, daratumumab, and venetoclax (Bcl-2 inhibitor) [118,150]. A phase Ib KEYNOTE-013 trial (NCT01953692) for pembrolizumab, a PD-1 inhibitor, in RRMM patients showed 2.7 months PFS and 20.2 months OS after a median follow-up of 19.9 months [151]. In 2017, the FDA terminated two phase III trials, KEYNOTE-183 (NCT02576977), and KEYNOTE-185 (NCT02579863), in which pembrolizumab was tested in RRMM and NDMM patients with pomalidomide or lenalidomide, respectively, in addition to a low dose of dexamethasone. In the trials, there was an increase of progression risk and death as the PFS in the pembrolizumab-Pd arm was shorter than in the Pd arm [151]. 

Siltuximab, an anti-IL-6 monoclonal antibody, was not effective when tested alone or in combination with dexamethasone in a in a phase II clinical trial, NCT00911859 [152]. Similarly, another phase II clinical study showed that the combination of siltuximab with bortezomib did not improve PFS or OS of RRMM patients [153]. However, when testing siltuximab on patients with SMM, there was a delay in the progression of high-risk SMM [154].

### 3.2. Antibody–Drug Conjugate

To enhance specificity of cytocidal compounds and reduce their off-target effects, an engineered monoclonal antibody is used as a carrier directed toward a tumor-associated antigen. Antibody–drug conjugate (ADC) is one of the exciting and fast evolving approaches in immunotherapy that enhances specificity. This engineered antibody is conjugated with the cytotoxic agent (the payload) through a linker, which should be stable in circulation to guarantee the attachment of the toxic payload until the antibody gets internalized inside the cell (Figure 6E) [155].

Belantamab mafodotin was the first-in-class monoclonal antibody that was approved to treat MM [156]. It is an anti-BCMA that is covalently linked to monomethyl auristatin F, a microtubule inhibitor [157]. There are some other ADCs that have been developed as anti-BCMA drug conjugates; for example, HDP-101, an anti-BCMA drug conjugate, has been tested in preclinical trials. It is conjugated to alpha-amanitin, a eukaryotic RNA polymerase II inhibitor, to inhibit transcription and therefore translation [158,159,160].

### 3.3. Bispecific Antibodies

Bispecific antibodies are monoclonal antibodies that have two binding sites for different antigens or two different epitopes on the same antigen (Figure 6D). They work through four different mechanisms: (1) bind target cell and an immune cell to assist immune response, (2) block two signaling pathways to prevent immune escape of cancerous cells, (3) block two immune checkpoints, and (4) drive connection of protein complexes [161]. The most common mechanism known for bispecific antibodies is to simultaneously bind two different antigens on two different cells such as a cancer cell and a T-cell to bring them into proximity and facilitate antitumor immune response [162]. 

In 2022, the FDA approved Teclistamab as the first bispecific antibody to treat MM patients who had received at least four prior treatments. It targets both CD3 on T-cells and BCMA, which is overexpressed on MM cells. Bringing T-cell and MM cells in proximity creates an immunological synapse and initiates cytolytic cascades and stimulates proinflammatory cytokines. MM cell death and lysis occur after T-cell activation. Teclistamab was approved after a phase I/II MajesTEC-1 trial (NCT04557098) in triple-class RRMM with five previous therapy lines. The ORR was 63.0% at a median follow-up of 14.1 months with 39.4% of patients showing complete response; PFS was 11.3 months. The common adverse effects are cytokine release syndrome, thrombocytopenia, neutropenia, and anemia. Additionally, 14% of patients had neurotoxic events. Currently, a phase III trial (NCT05083169), which is still recruiting, will be testing the combination of teclistamab with daratumumab (Tec-Dara) versus DPd (daratumumab, pomalidomide, and dexamethasone) or DVd (daratumumab, bortezomib, dexamethasone) in RRMM [140,141]. 

Elranatamab, another bispecific antibody, also targets CD3 and BCMA and is now in a phase III trial for RRMM. It has been tested as monotherapy as well as in combination in Magnetis MM clinical trials. It showed a good ORR (around 60%) and it was well tolerated in patients [163]. Talquetamab, is a bispecific antibody that targets both CD3 and the orphan G protein coupled receptor, class C group 5 member D (GPRC5D) that has been shown to be expressed suggested as tumor load in MM patients [164,165].

### 3.4. Chimeric Antigen Receptor-Modified T-Cells and NK Cells

CARs are genetically engineered transmembrane receptors designed to recognize and target specific antigens on cell surfaces. CAR-T cells are generally engineered in vitro after collecting autologous patients’ T-cells and transducing them with engineered lentivirus (Figure 6A). With CAR therapy, the immune system of patients can be reprogrammed and directed to target cancer cells [151]. CARs in general are composed of an extracellular domain that recognizes antigens such as single-chain variable fragment (scFv), and an intracellular activation domain (Figure 6B). The first generation of CAR, which was engineered in 1992, contains only CD3ζ. Then, one costimulatory domain, CD28 or 4-1BB (second generation), or two costimulatory domains (third generation), were added to enhance CAR function [123]. After in vitro engineering, CAR cells are reinfused in patients’ bodies so that they would recognize and attach to specific antigens on tumor cells (Figure 6C). This engagement stimulates the signaling cascade in engineered cells. In the case of engineered T-cells, for example, engagement of CAR-T with its target can stimulate different signaling pathways such as the PI3K and MAPK pathways, which activate T-cells to release pro-inflammatory cytokines such as IFN-γ, TNF-α, IL-6, and IL-2 [123,166]. This activation of CAR-T cells is required to exert cytotoxic function and lysis of tumor cells. However, immune-mediated adverse reactions such as cytokine release syndrome (CRS) is one of the issues that researchers try to ameliorate by using different methods such as introducing “suicide” genes [167]. The field of CAR engineering to design next generation therapies is rapidly moving forward. The fourth generation of CAR is called T-cells redirected for antigen-unrestricted cytokine-initiated killing, in which a constitutively expressed chemokine is added to the second generation of CARs. When the CAR is activated, the cytokine is released to induce tumor killing [168]. To add a third synergistic signal to CD3ζ CD28, a truncated IL-2 receptor is added with a binding site for STAT3 and integrated into the second generation of CARs to produce the fifth generation of CARs [169]. Researchers are improving the CAR-T design by adding other targets in the same CAR-T construct and testing other costimulatory domains as well as reprogramming several types of immune cells.

The first clinical trial of an anti-BCMA CAR-T (Construct: anti-BCMA scFv, CD8α hinge and transmembrane regions, the cytoplasmic portion of the CD28 costimulatory moiety, and the CD3ζ T-cell activation domain) started in 2014 (NCT02215967) after conditioning with cyclophosphamide and fludarabine. Antimyeloma activity and remissions of poor prognosis RRMM patients have been reported; however, side effects of CRS were seen in two patients out of eleven. Moreover, antigen escape has been seen in patients treated with anti-BCMA CAR-T [170,171].

Various anti-BCMA CAR-T therapies have been developed and investigated in clinical trials (Table 4). One of the challenges facing immunotherapy targeting antigens is the antigen escape that has been reported in many targeted antigens, including BCMA, which leads to relapse of MM patients and shorter durations of remission [172]. For this reason, researchers are attempting to find a ligand that can bind with more than one antigen. The BAFF ligand binds with three receptors on mature B cells: BAFF-R, BCMA, and transmembrane activator and calcium-modulating cyclophilin ligand interactor (TACI). Therefore, Wong et al. constructed a BAAF ligand-CAR-T cell (Construct: extracellular BAFF ligand, short spacer, hinge from human IgG1, CD28 transmembrane and signaling domains, OX40, and CD3ζ), which can interact with all three proteins (BAFF-R, BCMA, and TACI). BAAF L-CAR-T was significantly activated when co-cultured with U266, RPMI8226, or MM.1S MM cells. Furthermore, BAAF L-CAR-T showed a significant cytotoxic effects in vivo using xenograft models injected with MM1.S [173]. A phase I clinical trial is ongoing, which will study the efficacy of BAAF ligand-CAR-T in MM patients (NCT05546723). 

Currently, idecabtagene (BCMA CAR-T) and ciltacabtagene autoleucel are the only CAR-Ts approved to treat RRMM. Idecabtagene vicleucel (ide-cel; bb2121) was approved in March 2021 after the phase II KarMMa trial (NCT03361748) on RRMM patients who received ≥3 prior regimens. The ORR was 73.0% at a median follow-up of 11.3 months with PFS around 8.6 months. The common adverse effects are CRS, cytopenias, and neurotoxicity [135]. Likewise, ciltacabtagene autoleucel (cilta-cel), which contains two anti-BCMA-targeting single domains was approved in 2022 after a phase Ib/II CARTITUDE-1 trial (NCT03548207) on double refractory MM patients who received four or more prior lines of therapy, including a PI, an IMiD, and an anti-CD38 monoclonal antibody. The ORR was 100% at a median follow-up of 9 months. PFS was 6 months with 76% CR and 21% had VGPR. The common adverse effects are CRS, neutropenia, thrombocytopenia, and leukopenia [139].

Even though anti-BCMA CAR-T showed promising results in MM clinical trials, there are concerns about toxicity, CRS, and aplasia [174,175]. Therefore, NK cells could be better candidates [176]. CAR-NK trafficking of CAR-NK cells to the BM is challenging. To solve this issue, Yu et al. have an exciting in vivo study with a xenograft mouse model in which they modified anti-BMCA CAR-NK cells to express CXCR4 to enable the CAR-NK infiltration in the BM. For the CAR-NK construct, they used BCMA-specific scFv binding domain and compared two intracellular activation domains: CD3ζ and DAP12. There was no statistically significant difference in the performance of CAR-NK between the two activation domains. Expressing CXCR4 in NK cells significantly increased migration towards SDF-1α in vitro and promoted their migration toward BM in vivo. Moreover, expressing CXCR4 in anti-BCMA CAR-NK significantly inhibited tumor burden after 42 days [177]. An early phase I trial started in June 2021 and the final data was expected to be collected in December 2022 (NCT04727008). Other clinical trials for anti-BMCA CAR-NK are still recruiting patients (Table 4).

## 4. Glance into the Future

Even though the MM survival rate has improved with current treatments, generally 19–25% patients do not respond to PIs during the first-time treatment and almost 50% of RRMM patients do not respond to PIs, which leads to a serious problem [118]. The OS of RRMM patients is about 8 months [46,118]. Therefore, it is necessary to find new treatments and combination strategies to overcome drug resistance and improve survival for MM patients. Several new targets have been identified, and new molecules have been developed. Some of these molecules have been introduced in the clinical trial and some are still in the pre-clinical phase. 

### 4.1. Targeting the Apoptotic Pathway in MM

Apoptosis, a regulated cell death, is an essential mechanism in regulating development and maintaining homeostasis, plays a crucial role in preventing oncogenesis. The unrestrained growth of MM can occur due to the loss of control of apoptosis that happens through upregulation of antiapoptotic proteins such as Mcl-1, Bcl-2, and Bcl-xL all of which protect against genomic instability. Activation of different signaling cascades such as JAK2/STAT3, NF-κB, PI3K/AKT/mTOR, and Wnt/β-catenin pathways through the tumor microenvironment leads to upregulation of antiapoptotic proteins.

The efforts of targeting different molecules in apoptotic and survival pathways has shown promising results. Many small molecules that can inhibit survival and/or stimulate apoptosis have been developed, some of which are still in clinical trials. 

#### Bcl-2 Family Inhibitors

The Bcl-2 family is one of the most important targets, and their inhibitors are either in clinical or in preclinical stages. For example, venetoclax, a Bcl-2 inhibitor, is approved for the treatment of hematological malignancies including chronic lymphocytic leukemia, AML, and small lymphocytic lymphoma [178,179]. There are around 25 clinical trials for venetoclax with different drug combinations ongoing in MM [180]. In a phase III BELLINI clinical trial (NCT02755597) for RRMM patients, a significant improvement in PFS could not be achieved when venetoclax was used in combination with bortezomib and dexamethasone [181]. However, the study showed that venetoclax is more effective in patients with t (11;14) and high Bcl-2 expression compared to the other patients [181]. 

The overexpression of another Bcl-2 family member, Mcl-1, is reported in MM patients. The expression of Mcl-1 is associated with 1q21 [182] amplification, where approximately 40% of ND and 70% of RR patients show gain (3 copies) or amplification (≥4 copies) in 1q21 [183,184]. The overexpression of Mcl-1 is associated with relapse and poor prognosis [185]. Moreover, Mcl-1 dependent cancers are resistant to pan Bcl-2/Bcl-xL inhibitors (ABT-737), and venetoclax [186]. Therefore, Mcl-1 is an attractive target for MM [187,188,189]. There are several clinical trials ongoing for Mcl-1 inhibitors, for example MIK665 (also named S64315) is tested in phase I trials for refractory or relapsed lymphoma or MM patients [190]. One of these trials (NCT02992483) is completed, but results are not yet released [191]. The other trial (NCT04702425) is still recruiting with the goal of testing the combination of MIK665 with a Bcl-2 inhibitor, VOB560 [191]. Both in vitro and in vivo studies have shown that the combination of Mcl-1 inhibitor, MIK665, with Bcl-2 inhibitors showed strong and durable antitumor responses [192,193]. Other potential Mcl-1 inhibitors, PRT14 and AMG176 (Tapotoclax), are currently in phase I (NCT04543305, NCT02675452).

Marinopyrrole A (also named maritoclax), a natural product from marine-derived streptomycetes was discovered by the Wang group as a selective Mcl-1 antagonist [194]. They found that Maritoclax induces caspase-3 activation by directly binding to Mcl-1 and targeting it for proteasomal degradation and sensitizes cancer cells to ABT-737. The researchers showed that maritoclax disrupts the interaction between Bim and Mcl-1 [194]. Along these lines, we showed that maritoclax potentiates the apoptotic effect of ABT-737 in human melanoma cells [195].

### 4.2. Targeting MM Cancer Stem Cells

It has been demonstrated that cancer stem cells (CSCs) are critical in relapse [196]. The mechanism of how CSCs are involved in relapse is not well understood; however, it has been demonstrated that intracellular drug detoxification and drug efflux which retribute to the overexpression of aldehyde dehydrogenase (ALDH) and ATP-binding cassette transporter G2 (ABCG2) are key players [197]. Several studies have been made to investigate phenotypic characteristics and identify surface markers for CSCs. However, the molecular profile of CSCs is still controversial. The studies have demonstrated that CD138^−^ PCs exhibit tumor-initiating potential, self-renewal capability, and drug resistance, which suggests that CD138^−^ is a CSC [198]. Furthermore, it is likely that because of the heterogenous characteristic of MM, there are different stem cell subsets [199]. For example, the markers CD19^+^CD38^−^CD27^+^, CD19^+^CD34^−^Lchain(λ)^+^ALDH^+^, and CD19^−^CD45^−^CD38^+^CD138^+^ have also been identified as CSCs in MM [200].

Some natural compounds have been shown to be effective in targeting CSCs. The ethanolic extract of scutellaria, a traditional Chinese herbal remedy, and its derivatives baicalein, wogonin, and baicalin have been shown to decrease the expression level of ABCG2 protein in RPMI-8226 [201].

Salinomycin, a monocarboxylic polyether antibiotic derived from *Streptomyces albus* [202], is shown to inhibit stemness in cancer cells. It overcomes ABC transporter-mediated multidrug resistance and induces apoptosis in human leukemia stem-cell-like cells [203]. Moreover, Kastritis et al. showed that salinomycin decreases the side population fraction of MM, which is known to efflux Hoechst stain and represent a stem-cell-like population [204].

ALDHs are detoxifying enzymes that have been shown to be highly expressed in HSCs and CSCs [205]. The overexpression of ALDH in cancer has been associated with drug resistance, relapse, and poor prognosis [206]. Zhou et al. showed that ALDH is associated with chromosomal instability in MM and that an ALDH1+ subset from two MM cell lines had a higher clonogenic potential than an ALDH1^−^ cell subset [207]. Moreover, Yang et al. found that overexpression of ALDH1 increased MM drug resistance in vivo [208]. Even though ALDH inhibitors are not in the clinical trials for MM, it is a compelling target to explore, and researchers are developing and modifying different ALDH inhibitors (reviewed in [209]).

### 4.3. Targeting the Bone Marrow Microenvironment

The interaction of myeloma cells to BMM is a hallmark of MM. This interaction supports myeloma cell survival and plays critical role in pathogenesis, thus targeting of BMM has been one of the areas of active research. Along these lines, SDF-1α/CXCR4 axis, Bruton’s tyrosine kinase (BTK), JAK/STAT, NF-κB, and RANK/RANK-L have been identified as prime targets. 

Plerixafor, a specific antagonist of SDF-1α binding to CXCR4, was approved in 2008 to induce HSCs and human progenitor cells (HPCs) trafficking. It has been shown that it augments granulocyte colony-stimulating factor (G-CSF)-induced mobilization of HSCs and HPCs [38,39]. Our lab has shown that gambogic acid, a xanthonoid derived from *Garcinia hanburyi*, blocks RANKL-induced osteoclastogenesis, suppressing the SDF-1α- induced chemotaxis of MM cells [210].

#### Bruton’s Tyrosine Kinase Inhibitors

Kinases play essential roles in survival pathways; therefore, a wide range of kinase inhibitors have been developed and investigated regarding MM. There are numerous potential targets for kinase inhibitors, including but not limited to non-receptor tyrosine kinases, receptor tyrosine kinases (RTKs), PI3K/AKT/mTOR pathway kinases, protein kinase C, mitogen-activated protein kinase, cell cycle control kinases, casein kinase, and glycogen synthase kinase. At this writing, there are about 100 clinical trials testing different kinase inhibitors in the targeting of MM survival pathways.

One of the essential kinases that is important during B-cell development is BTK, which belongs to the Tec family of kinases. BTK is shown to be overexpressed and activated in MM stem-cell-like cells [211]. BTK is activated by different pathways including B-cell receptor (BCR), toll-like receptor, chemokine, and Fc receptor signaling pathways [212]. When activated, it translocates from the cytosol to the cell membrane and gets phosphorylated by one of the members of the spleen tyrosine kinase (SYK) or Src family kinases on the tyrosine residue at position 551 (Y551) [213]. Then, an autophosphorylation occurs on tyrosine at position 223 (Y223). The activation of BTK stimulates several downstream signals that are associated with survival, proliferation, drug resistance, and migration. 

Ibrutinib and acalabrutinib are FDA-approved BTK inhibitors to treat B-cell malignancies other than MM. Ibrutinib inhibits BTK by covalently binding cysteine at position 481 (C481). BTK inhibitors are not yet approved for MM; however, clinical trials are ongoing (Table 2). After the excitement of the effective results of ibrutinib and acalabrutinib in treating B-cell malignancies, resistance occurred due to acquired mutations in the kinase domain such as the Cys481 to Ser (C481S) mutation, which disrupts the covalent binding of these drugs. Therefore, we developed a new BTK inhibitor, KS151, that avoids binding with the Cys481 residue. Importantly, this new molecule was effective in a stem-cell-like population and kills MM cells [214]. 

## 5. Conclusions

Most MM patients experience a relapse after treatment which can occur from drug resistance and antigen escape. The genomic instability that is associated with the disease progression adds to the complexity of disease progression. Moreover, there are multiple mutational drivers that can lead to MM disease and contribute to its heterogeneity [109,215]. Misund et al. showed that MM progression is associated with phenotypic transformation and several changes in the transcriptomic levels in patients’ sample which can be targeted in future [216]. There are various pathways that are disrupted in MM that can affect other pathways by cross talk activation or repression. Therefore, it is essential to combine different treatments to overcome different feedback loops that can counteract the effect of some inhibitors, overcome resistance, and lower the side effects of given medications.

The treatment of MM continues to progress, and the future of immunotherapy in combination with other treatments will focus on understanding the MM stem-cell-like cells and finding putative tumor-associated antigens that can be targeted with immunotherapy as well as small molecules, improving the efficacy and the specificity of immunotherapy, and targeting the BMM compartments such as BMSCs to overcome the BMM’s protective niche. Over the past decade, immunotherapy has been introduced and has shown promising results. There is a need to have more clinical studies regarding treatment sequences to know the effect of immunotherapy on NDMM patients. Personalized treatments should also be considered as MM has a heterogeneous phenotype. New targets should be investigated with novel compounds to minimize toxicity and side effects and increase patients’ OS and quality of life.

## Figures and Tables

**Figure 1 pharmaceuticals-16-00415-f001:**
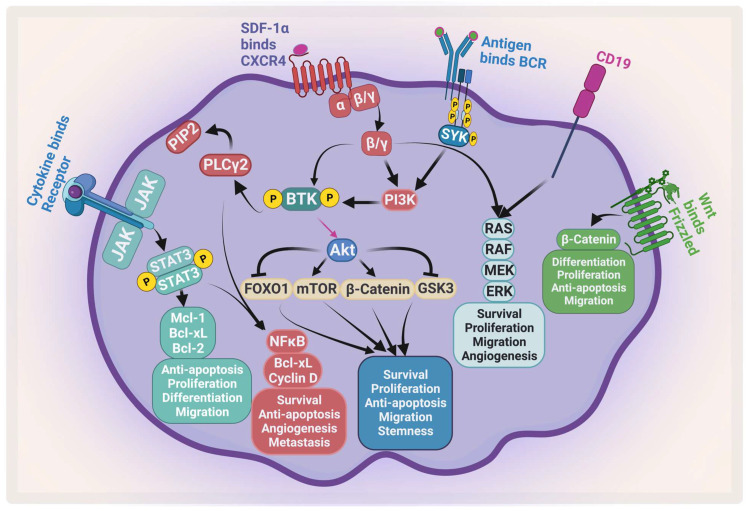
**Major signaling pathways in MM**. MM receives several survival and proliferation signals: JAK2/STAT3 pathway activates antiapoptotic proteins and activates NF-κB, the PI3K/AKT/mTOR pathways become activated when stromal-derived factor 1 α (SDF-1α) binds to CXCR4 and/or antigen binds to B-cell receptor (BCR). The RAS/MEK/ERK pathway becomes activated by BCR and CD19. Wnt/β-catenin pathway activation enhances differentiation, survival, migration, and antiapoptotic signals. Ras: rat sarcoma; Raf: rapidly accelerated fibrosarcoma; MEK: mitogen-activated protein kinase kinase. Created by Biorender.com.

**Figure 2 pharmaceuticals-16-00415-f002:**
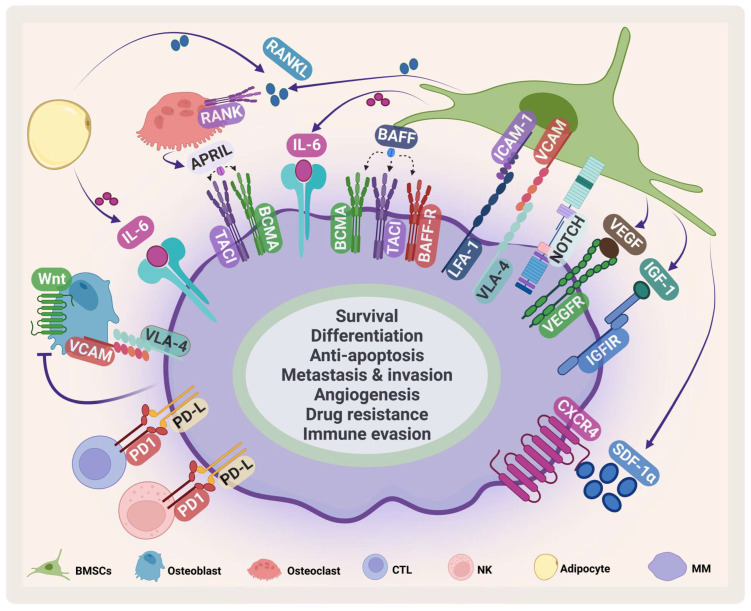
**MM microenvironment**. MM cells interact with several cells in the BMM such as BMSCs, osteoclasts, osteoblasts, cytotoxic T-cell (CTLs), natural killer (NK) cells, adipocytes. This interaction enhances MM survival and inhibits immune cells. VCAM: vascular cell adhesion protein; LFA-1: lymphocyte function-associated antigen-1; ICAM-1: intercellular adhesion molecule 1; IGF-1: insulin-like growth factor-1. Created by Biorender.com.

**Figure 3 pharmaceuticals-16-00415-f003:**
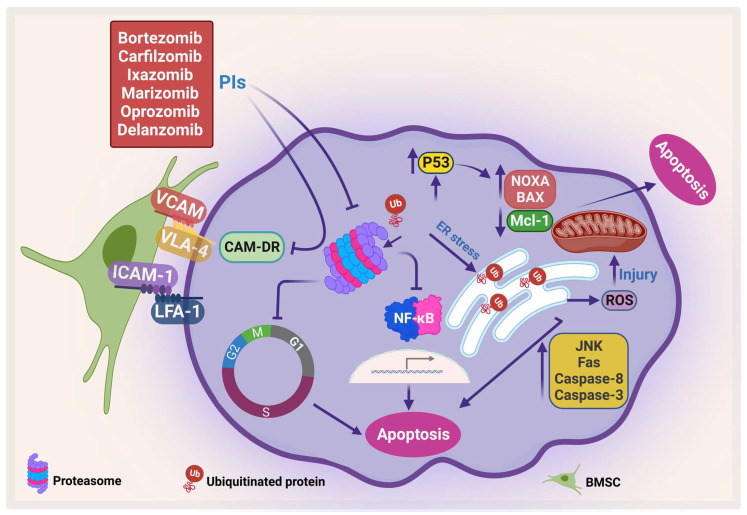
**Proteasome inhibitors (PIs) induce apoptosis and inhibit CAM-DR**. PIs induce protein accumulation, which leads to ER stress and activates JNK, which in turn activates caspase-8 and caspase-3, increases Fas, and generates ROS. In addition, PI enhances pro-apoptotic proteins, NOXA and BAX, and inhibits Mcl-1, which leads to apoptosis. Moreover, PI inhibits CAM-DR through inhibition of adhesion molecules (adapted from [86]). Created by Biorender.com.

**Figure 4 pharmaceuticals-16-00415-f004:**
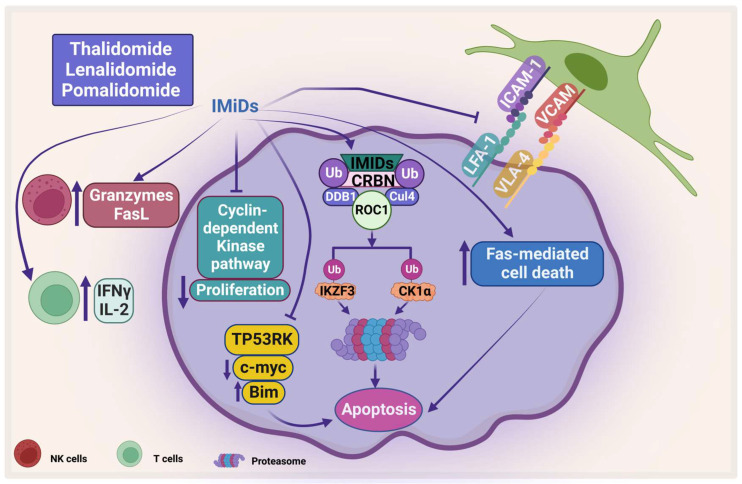
**The direct and indirect effect of IMiDs on MM**. IMiDs enhances apoptosis and inhibits MM proliferation through the cyclin-dependent pathway. IMiDs works indirectly through activating the immune cells. CRBN: Cereblon, DDB1: DNA damage-binding protein 1, CUL4: Cullin-4A, and ROC1: RING box protein-1. Created by Biorender.com.

**Figure 5 pharmaceuticals-16-00415-f005:**
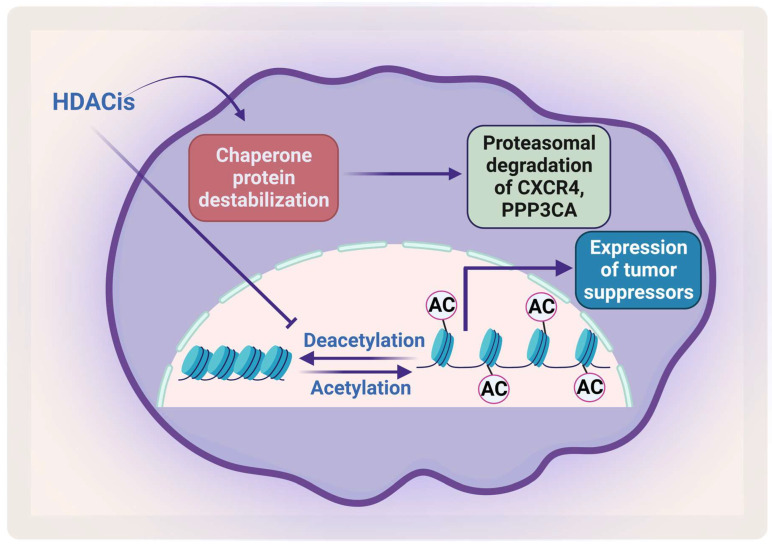
**The effect of HDACis on MM**. HDACis cause proteasomal degradation of CXCR4 and PPP3CA and enhance the expression of tumor suppressors. Created by Biorender.com.

**Figure 6 pharmaceuticals-16-00415-f006:**
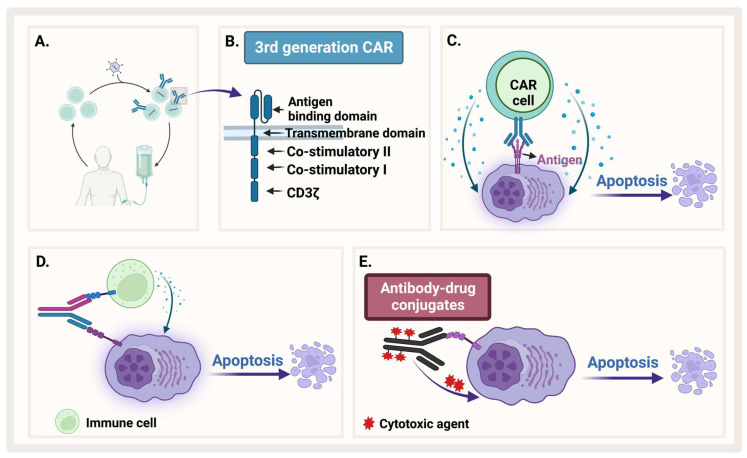
**Immunotherapy**. (**A**) Generation of CAR-T. (**B**) The different generations of CAR-T. (**C**) CAR-T recognizes an antigen on MM and causes apoptosis. (**D**) Bispecific antibody binds both MM and an immune cell. (**E**) Antibody–drug conjugate recognizes MM and releases cytotoxic agent. Created by Biorender.com.

**Table 1 pharmaceuticals-16-00415-t001:** FDA-approved medications for MM from 2006 to 2013.

Drug	Year	Treatment	Adverse Effects	Refs.
**Thalidomide**	2006	NDMM	somnolence, constipation, neuropathy, VTE, and rash	[48,49]
**Lenalidomide**	2006	Received one prior therapy	neutropenia, thrombocytopenia, leukopenia, lymphopenia, febrile neutropenia, deep vein thrombosis, pulmonary embolism, atrial fibrillation, constipation, diarrhea, fatigue, pneumonia, hypokalemia, hypocalcemia, muscle weakness, neuropathy, and depression	[50]
**Doxorubicin**	2007	RRMM	thrombocytopenia, neutropenia, anemia, fatigue, pyrexia, nausea, vomiting, mucositis/stomatitis diarrhea, and hand foot syndrome	[49,51]
**Bortezomib**	2008	NDMM	asthenic conditions, diarrhea, constipation, PSN, vomiting, nausea psychiatric disorders, pyrexia, anorexia, thrombocytopenia, leukopenia, neuralgia, neutropenia, and anemia	[49,52]
**Plerixafor**	2008	MM	diarrhea, vomiting, nausea, fatigue, headache, injection site reactions, dizziness, and arthralgia	[49,53]
**Carfilzomib**	2012	RRMM	fatigue, anemia, nausea, thrombocytopenia, dyspnea, diarrhea, pyrexia, pneumonia, ARF, pyrexia, and CHF	[54]
**Pomalidomide**	2013	RRMM	asthenia, fatigue, neutropenia, anemia, constipation, diarrhea, nausea, URTI, dyspnea, back pain, and pyrexia	[55]

**NDMM**: newly diagnosed multiple myeloma; **RRMM**: relapsed/refractory multiple myeloma; **NTE**: non-transplant-eligible; **TE**: transplant eligible; **Td**: thalidomide, dexamethasone; **Rd**: lenalidomide, dexamethasone; **V**: bortezomib; **PLD-V**: pegylated liposomal doxorubicin, bortezomib; **MP**: melphalan, prednisone; **VMP**: bortezomib, melphalan, prednisone; **G-CSF**: granulocyte-colony-stimulating factor; **Plerix-G-CSF**: plerixafor, granulocyte-colony-stimulating factor; **Pd**: pomalidomide, dexamethasone; **VTE**: venous thromboembolism; **PSN**: Peripheral sensory neuropathy; **ARF:** acute renal failure; **CHF**: congestive heart failure; **URTI**: Upper respiratory tract infection.

**Table 2 pharmaceuticals-16-00415-t002:** FDA-approved medications for MM from 2014 to 2019.

Drug	Year	Treatment	Adverse Effects	Name and NCT Number	Refs.
**Panobinostat**	2015	RRMM	pneumonia, diarrhea, arrhythmias, hypophosphatemia and hypokalemia, ECG change, thrombocytopenia, neutropenia fatigue, and sepsis	PANORAMA1 NCT01023308	[69]
**Carfilzomib**	2015	RRMM	CVE, VTE, ARF, pulmonary toxicities, hypertension, and thrombocytopenia	ASPIRE NCT01080391	[70]
**Daratumumab**	2015	RRMM	fatigue, nausea, back pain, pyrexia, URTI, cough, IRs, lymphopenia, neutropenia, anemia, and thrombocytopenia	SIRIUS NCT01985126	[71]
**Ixazomib**	2015	RRMM	diarrhea, constipation, thrombocytopenia, PSN, nausea, peripheral edema, vomiting, and back pain	TOURMALITOURMALINE NCT01564537	[47]
**Elotuzumab**	2015	RRMM	ARF, pneumonia, nasopharyngitis pyrexia, anemia, pulmonary embolism, and PSN	ELOQUENT-2 NCT01239797	[72] [73]
**Daratumumab**	2016	RRMM	URTI, cough, diarrhea, fatigue, nausea, pyrexia, muscle spasm, and dyspnea, neutropenia, anemia	POLLUX NCT02076009	[74]
**Daratumumab**	2016	RRMM	URTI, IRs, diarrhea, peripheral edema, Neutropenia, and thrombocytopenia, anemia	CASTOR NCT02136134	[75,76]
**Daratumumab**	2019	NTE NDMM	IRs, URTI, diarrhea, constipation, peripheral edema, nausea, fatigue, asthenia, dyspnea, pyrexia, muscle spasms, and PSN	MAIA NTC02252172	[66,77,78]
**Selinexor**	2019	RRMM	Thrombocytopenia, fatigue, nausea, anemia, diarrhea, vomiting, hyponatremia, neutropenia, leukopenia, constipation, dyspnea, and URTI	STORM KCP-330-012 NCT02336815	[79,80]
**Daratumumab**	2019	TE NDMM	IRs, PSN, constipation, asthenia, nausea, neutropenia, thrombocytopenia, peripheral edema, pyrexia and paresthesia	CASSIOPEIA NCT02541383	[81]

**NDMM**: newly diagnosed multiple myeloma; **RRMM**: relapsed/refractory multiple myeloma; **NTE**: non-transplant-eligible; **TE**: transplant eligible; **ORR**: overall response rate; **Vd:** bortezomib, dexamethasone; **PAN-Vd:** panobinostat, bortezomib, dexamethasone; **Rd**: lenalidomide, dexamethasone; **KRd:** carfilzomib, lenalidomide, dexamethasone; **DVd:** daratumumab, bortezomib, dexamethasone; **DRd**: daratumumab, lenalidomide, dexamethasone; **IRd:** ixazomib, lenalidomide + dexamethasone; **VTd**: bortezomib, thalidomide, and dexamethasone; **DVTd:** daratumumab, bortezomib, thalidomide, dexamethasone; **ERd**: elotuzumab, lenalidomide, dexamethasone; **Sd**: selinexor, dexamethasone; **m**: months; **PFS**: progression-free survival; **NR**: Not Reached; **URTI:** Upper respiratory tract infection; **PSN**: Peripheral sensory neuropathy; **IRs**: Infusion reactions; **VTE:** venous thromboembolic events; **CVE**: Cardiovascular events, **ARF**: acute renal failure.

**Table 3 pharmaceuticals-16-00415-t003:** FDA-approved medications for MM from 2017 to 2022.

Drug	Year	Treatment	Adverse Effects	Name & NCT Number	Refs.
**Isatuximab**	2020	RRMM	neutropenia, IRs, pneumonia, URTI, and diarrhea	ICARIA-MM NCT02990338	[62]
**Daratumumab**	2020	RRMM	URTI, constipation, nausea, fatigue, pyrexia, PSN, diarrhea, cough, insomnia, vomiting, back pain, muscle spasms, dyspnea, neutropenia, thrombocytopenia, and anemia	COLUMBA NCT03277105	[129]
**Belantamab mafodotin**	2020	RRMM	keratopathy, decreased visual acuity, nausea, blurred vision, pyrexia, IRs, and fatigue	DREAMM-2 NCT 03525678	[130,131]
**Carfilzomib**	2020	RRMM	IRs, anemia, diarrhea, fatigue, hypertension, pyrexia, respiratory tract infection, thrombocytopenia, anemia, neutropenia, lymphopenia, cough, dyspnea, insomnia, hypertension, headache, and back pain	EQUULEUS NCT01998971CANDOR NCT03158688	[132]
**Selinexor**	2020	RRMM	Nausea, fatigue, decreased appetite, diarrhea, PSN, URTI, decreased weight, cataract, vomiting, thrombocytopenia, lymphopenia, hypophosphatemia, anemia, hyponatremia, and neutropenia.	BOSTON NCT03110562	[78,133]
**Melphalan flufenamide**	2021	RRMM	fatigue, nausea, diarrhea, pyrexia, neutropenia, thrombocytopenia, anemia, and pneumonia	HORIZON NCT02963493	[134]
**Idecabtagene vicleucel**	2021	RRMM	CRS, neurologic toxicities, hemophagocytic lymphohistiocytosis/macrophage activation syndrome, prolonged cytopenias, infections, fatigue, musculoskeletal pain, and hypogammaglobulinemia.	KarMMa NCT02658929	[135]
**Isatuximab**	2021	NTE NDMM	URTI, bronchitis, cough, diarrhea, IRs, fatigue, hypertension, thrombocytopenia, and anemia	IKEMA NCT03275285	[82,136]
**Daratumumab**	2021	RRMM	IRs, fatigue, pneumonia, upper respiratory tract infection, and diarrhea, neutropenia, thrombocytopenia, anemia, and hyperglycemia	APOLLO NCT03180736	[137]
**Daratumumab**	2021	RRMM	URTI, hypertension, diarrhea, cough, fatigue, insomnia, pyrexia, nausea, and peripheral edema	PLEIADES NCT03412565	[138]
**Ciltacabtagene autoleucel**	2022	RRMM	pyrexia, cytokine release syndrome, hypogammaglobulinemia, musculoskeletal pain, fatigue, infections, diarrhea, nausea, encephalopathy, headache, coagulopathy, constipation, and vomiting	CARTITUDE-1 NCT03548207	[139]
**Teclistamab-cqyv**	2022	RRMM	CRS, ICANS, fatigue, pneumonia, diarrhea, pyrexia, neutropenia, thrombocytopenia, and anemia	MajesTEC-1 NCT0314518, NCT04557098	[140,141]

**NDMM**: newly diagnosed multiple myeloma; **RRMM**: relapsed/refractory multiple myeloma; **IV**: intravenous; **SC**: subcutaneous; **Pd**: pomalidomide, dexamethasone; **DPd**: daratumumab, pomalidomide, dexamethasone; **Isa-Pd**: isatuximab, pomalidomide, dexamethasone; **Kd**: carfilzomib, dexamethasone; **DKd**: daratumumab, carfilzomib, dexamethasone; **Isa-Kd**: isatuximab, carfilzomib, dexamethasone; **Vd**: bortezomib, dexamethasone; **SVd**: selinexor, bortezomib, dexamethasone; **ORR**: overall response rate; **VGPR**: very good partial response; **NR**: Not Reached; **IRs**: Infusion reactions; **CRS**: cytokine release syndrome; **ICANS**: immune effector cell-associated neurotoxicity; **URTI**: Upper respiratory tract infection; **PSN**: peripheral sensory neuropathy; **ARF**: acute renal failure.

**Table 4 pharmaceuticals-16-00415-t004:** Some of the ongoing CAR-T and CAR-NK trials.

Name/Target	Combination	Patients’ Status	Trial Number	Phase	Recruiting Status
**Anti-BCMA/GPRC5D**		RRMM	NCT05509530	Phase II	Recruiting
**APRIL CAR-T cells**		BCMA/TACI Positive RRMM	NCT04657861	Early Phase I	Recruiting
**Dual Specificity** **CD38 and BCMA**		RRMM	NCT03767751	Phase I	Unknown
**Anti-BCMA**	Fludarabine Cyclophosphamide	MM	NCT03322735	Phase I	Unknown
**SLAMF7 CAR-T**		MM	NCT04499339	Phase I/IIa	Recruiting
**CXCR4 modified** **anti-BCMA CAR T-cells**		MM	NCT04727008	Early Phase I	Not yet recruiting
**CD19-CD22 CAR-T-cells**		RRMM	NCT04714827	Phase I/II	Recruiting
**BCMA/CD19 Dual-Target CAR-T**		RRMM	NCT04182581	Early Phase I	Unknown
**CD 70 CAR T**		CD70 Positive RRMM	NCT04662294	Early Phase I	Recruiting
**Anti-CD38 CAR-T**		RRMM	NCT03464916	Phase I	Completed
**Bispecific CAR Targeting CS1 and BCMA**		RRMM	NCT03464916	Phase I	Completed
**Anti-BCMA**	clarithromycin, lenalidomide, dexamethasone	NDMM	NCT04287660	Phase III	Recruiting
**Anti-BCMA CAR-NK**		RRMM	NCT03940833	Phase I/II	Unknown
**Anti-BCMA CAR-NK**	Fludarabine Cytoxan	RRMM	NCT05008536	Early Phase I	Recruiting
**Anti-BCMA CAR-NK**	Cyclophosphamide Fludarabine Daratumumab	MM	NCT05182073	Phase I	Recruiting

**NDMM**: newly diagnosed multiple myeloma; **RRMM**: relapsed/refractory multiple myeloma.

## Data Availability

Data is contained within the article.

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
