# Peer review of "Past, Present, and a Glance into the Future of Multiple Myeloma Treatment"

_pharmaceuticals, 2023, doi:10.3390/ph16030415_

Round 1

Reviewer 1 Report

This review should focus (see abstract) on the current treatments of multiple myeloma and on the main pathways related to MM cells growth that could be targeted for future treatments. The paper is not well organized, it is often rather confusing and difficult to read (There are no Tables but they are mentioned in the text). There is a disproportion between the discussion about MM biology and novel possible targets and that on current therapies of MM that is just mentioned.  The topic “existing therapies for MM” is limited to a few lines and references are not always appropriate and often they are wrong (for example page 4 reference 48 the Authors could have cited Attal et al, NEJM 1996; reference 50 refers to VISTA trial and not to a Buda real life experience). The Authors did not mention the most recently approved regimens for newly diagnosed MM as DRd, D-VMP, D-VTD (see page 6 and 9) and trials leading to their approval as MAIA, ALCYONE, CASSIOPEIA (also in this case page 6 reference 60 is not appropriate). The paragraphs on bispespific and CAR T cell (pages 10, 11 and 12) are inadequate and, also in this case, the Author did not mention the main studies on these new therapies whereas they discuss too long about constructs or mechanisms of resistance.

See file Pharmaceuticals

Author Response

Comments: This review should focus (see abstract) on the current treatments of multiple myeloma and on the main pathways related to MM cells growth that could be targeted for future treatments. The paper is not well organized, it is often rather confusing and difficult to read (There are no Tables, but they are mentioned in the text). There is a disproportion between the discussion about MM biology and novel possible targets and that on current therapies of MM that is just mentioned.  The topic “existing therapies for MM” is limited to a few lines and references are not always appropriate and often they are wrong (for example page 4 reference 48 the Authors could have cited Attal et al, NEJM 1996; reference 50 refers to VISTA trial and not to a Buda real life experience). The Authors did not mention the most recently approved regimens for newly diagnosed MM as DRd, D-VMP, D-VTD (see page 6 and 9) and trials leading to their approval as MAIA, ALCYONE, CASSIOPEIA (also in this case page 6 reference 60 is not appropriate). The paragraphs on bispespific and CAR T cell (pages 10, 11 and 12) are inadequate and, also in this case, the Author did not mention the main studies on these new therapies whereas they discuss too long about constructs or mechanisms of resistance.

Response: Authors are thankful for reviewer’s critical evaluation and suggestions. Based on the comments we have revised the manuscript. We modified review in a way that it focuses on current treatment and main pathways that could be targeted for future therapies.  Specifically, we did following:

  1. Regarding organization of the review and lack of tables, we modified text for better flow and organization. Indeed, table was added during submission, somehow it was not converted in the final pdf. Please accept our apologies. Not only we added the tables, we modified also for better presentation.
  2. Regarding disproportion between the discussion about MM biology and novel possible targets, we modified in a way that it flows better and covers in-depth the MM biology and targets.
  3. Regarding limited explanation about existing therapies for MM, and wrong references, we thoroughly modified this section and added correct references.
  4. Regarding lack of information about recent approved regimen for newly diagnosed MM, we provided in depth information as well corrected the references.
  5. Regarding inadequate information on bispecific and CART cells, and incorrect references, we modified information along and replaced references. We also added this information in Table 4.

Reviewer 2 Report

In this manuscript, Elbezanti WO, et al. reviewed the important pathways for pathogenesis of multiple myeloma (MM) and the current treatment drugs for MM. I think this is interesting and useful contribution. But I think there are some problems as indicated below. I believe that the authors should revise the manuscript to publish in Pharmaceuticals.

Major point

1.     The authors did not introduce ixazomib and other proteasome inhibitors in the text. I think that ixazomib has an important role as a maintenance therapy for MM and should be explained in the text.

2.     The authors did not explain about the TP53RK pathways as the mechanism of action of IMiDs except for CRBN modulation. It is known that IMiDs, pomalidomide more potently than lenalidomide, bind and inhibit TP53RK and downstream p53 activity. This is one of the important mechanisms of action of IMiDs and should be explained. In addition, pomalidomide is also a very important drug in treatment for MM and should be discussed more precisely in the text.

Minor point

1.      The image quality of figures is little low. Can you change to better quality images?

2.      Page 13, line 10 in 3.2. Targeting MM Cancer Stem Cells section: ‘inhibts’ should be corrected to ‘inhibits’.

I hope that my comment is very useful for the improvement of the article.

Author Response

In this manuscript, Elbezanti WO, et al. reviewed the important pathways for pathogenesis of multiple myeloma (MM) and the current treatment drugs for MM. I think this is interesting and useful contribution. But I think there are some problems as indicated below. I believe that the authors should revise the manuscript to publish in Pharmaceuticals.

Response: We are thankful for reviewer’s encouraging remark. Based on the comments we modified manuscript. We appreciate that the comments tremendously improved the quality of manuscript.  

Major point

Comment 1: The authors did not introduce ixazomib and other proteasome inhibitors in the text. I think that ixazomib has an important role as a maintenance therapy for MM and should be explained in the text.

Response 1: We discussed about ixazomib and other proteasome inhibitors in text and accordingly we modified the table as well.

Comment 2: The authors did not explain about the TP53RK pathways as the mechanism of action of IMiDs except for CRBN modulation. It is known that IMiDs, pomalidomide more potently than lenalidomide, bind and inhibit TP53RK and downstream p53 activity. This is one of the important mechanisms of action of IMiDs and should be explained. In addition, pomalidomide is also a very important drug in treatment for MM and should be discussed more precisely in the text.

Response 2: As suggested we have now explained precisely about TP53RK pathways, and pomalidomide.

Minor point

Comment 3: The image quality of figures is little low. Can you change to better quality images?

Response 3: We replaced the images with better resolution.

Comment 4: Page 13, line 10 in 3.2. Targeting MM Cancer Stem Cells section: ‘inhibts’ should be corrected to ‘inhibits’. 

Response 4: It is corrected now. Thanks.

Comment 5: I hope that my comment is very useful for the improvement of the article.

Response 5: We are thankful for your precious comments. It certainly improved the manuscript.

Reviewer 3 Report

Pharmaceuticals-2167639

Past, Present and a Glance into the Future of Multiple Myeloma Treatment.  

The review article “Past, Present and a Glance into the Future of Multiple Myeloma Treatment. (Pharmaceuticals-2167639)” by Elbezanti WO, et al. summarized therapeutics for myeloma. This review article was organized well, especially figures were clear. However, there were one major issue and several minor issues for acceptance as below.

Major issue

Tables play important roles for reader to understand well. Therefore, I recommend that the authors add a table about ongoing or planed clinical trials about new therapeutics.

Minor issues

Which was he abbreviation for bone marrow microenvironment BMME or BMM? The authors should unify about that.

The author could add that HDAC6 inhibition works for anti-myeloma effect via ER stress with proteasome inhibitors synergistically.

The authors’ comment about daratumumab combination therapy was not clear. DARA combination therapy was various, and so “Then, it was approved to be used in com- bination with bortezomib and dexamethasone as well as in combination with lenalidomide and dexamethasone. Even though, it has a great effect on patients’ OS, a diminished response to treatment has been observed due to antigen loss (84). ” should be revised.

The author could explain that anti-CD38 monoclonal antibody works as an immunomodulator.

The author could add that elotuzumab have antimyeloma effect not only ADCC and ADCP but also reduction of soluble SLAMF7.

The author could add that clinical trials for anti-PD-1 monoclonal antibody, including both efficacy and safety.

The authors should show references for “Even though the MM survival rate has improved with current treatments,19% of patients do not respond to PI as the first line of treatment and 50% of relapsed MM patients do not respond to PI which leads to refractory situation.”

The author could add that MCL-1 inhibitor might be promising for 1q21 gain MM because MCL-1 was overexpressed in 1q21gain MM.

Author Response

The review article “Past, Present and a Glance into the Future of Multiple Myeloma Treatment. (Pharmaceuticals-2167639)” by Elbezanti WO, et al. summarized therapeutics for myeloma. This review article was organized well, especially figures were clear. However, there were one major issue and several minor issues for acceptance as below.

Response: Authors are thankful for reviewer’s comments and encouraging remark on figure.

Major issue

Comment 1: Tables play important roles for reader to understand well. Therefore, I recommend that the authors add a table about ongoing or planed clinical trials about new therapeutics.

Response 1: Authors agree that tables play important role in review articles. Indeed, we had several tables in previous version, somehow it did not appear in the main text. We have added tables now with ongoing and planned clinical trials.

Minor issues

Comment 2: Which was he abbreviation for bone marrow microenvironment BMME or BMM? The authors should unify about that.

Response 2: Thank you for suggestions, we modified the abbreviations throughout the text.  

Comment 3: The author could add that HDAC6 inhibition works for anti-myeloma effect via ER stress with proteasome inhibitors synergistically.

Response 3: As suggested we added about HDAC6 inhibitors and its synergy with PIs.

Comment 4: The authors’ comment about daratumumab combination therapy was not clear. DARA combination therapy was various, and so “Then, it was approved to be used in com- bination with bortezomib and dexamethasone as well as in combination with lenalidomide and dexamethasone. Even though, it has a great effect on patients’ OS, a diminished response to treatment has been observed due to antigen loss (84). ” should be revised.

Response 4: As suggested we modified the lines. 

Comment 5: The author could explain that anti-CD38 monoclonal antibody works as an immunomodulator.

Response 5: As suggested we explained about anti-CD138 as an immunomodulator.

Comment 6: The author could add that elotuzumab have antimyeloma effect not only ADCC and ADCP but also reduction of soluble SLAMF7.

Response 6: We added as advised.

Comment 7: The author could add that clinical trials for anti-PD-1 monoclonal antibody, including both efficacy and safety.

Response 7: We added now about anti-PD-1 and its efficacy.

Comment 8: The authors should show references for “Even though the MM survival rate has improved with current treatments,19% of patients do not respond to PI as the first line of treatment and 50% of relapsed MM patients do not respond to PI which leads to refractory situation.”

Response 8: We added reference for this sentence.

Comment 9: The author could add that MCL-1 inhibitor might be promising for 1q21 gain MM because MCL-1 was overexpressed in 1q21gain MM.

Response 9:  We added about 1q21 gain and Mcl-1i’s role.

Round 2

Reviewer 1 Report

no comment 

Reviewer 3 Report

This revised manuscript was organized well, and so I considered that this revised version was suitable for publication in the journal "Pharmaceuticals".